# Selective control of fcc and hcp crystal structures in Au–Ru solid-solution alloy nanoparticles

Quan Zhang[1], Kohei Kusada[1], Dongshuang Wu[1], Tomokazu Yamamoto[2,3], Takaaki Toriyama[3], Syo Matsumura[2,3,4], Shogo Kawaguchi[5], Yoshiki Kubota[6] & Hiroshi Kitagawa [1,4]

Binary solid-solution alloys generally adopt one of three principal crystal lattices—body-centred cubic (bcc), hexagonal close-packed (hcp) or face-centred cubic (fcc) structures—in which the structure is dominated by constituent elements and compositions. Therefore, it is a significant challenge to selectively control the crystal structure in alloys with a certain composition. Here, we propose an approach for the selective control of the crystal structure in solid-solution alloys by using a chemical reduction method. By precisely tuning the reduction speed of the metal precursors, we selectively control the crystal structure of alloy nanoparticles, and are able to selectively synthesize fcc and hcp $AuRu_3$ alloy nanoparticles at ambient conditions. This approach enables us to design alloy nanomaterials with the desired crystal structures to create innovative chemical and physical properties.

[1] Division of Chemistry, Graduate School of Science, Kyoto University, Kitashirakawa-Oiwakecho, Sakyo-ku, Kyoto 606-8502, Japan. [2] Department of Applied Quantum Physics and Nuclear Engineering, Kyushu University, 744 Motooka, Nishi-ku, Fukuoka 819-0395, Japan. [3] The Ultramicroscopy Research Center, Kyushu University, Motooka 744, Nishi-ku, Fukuoka 819-0395, Japan. [4] INAMORI Frontier Research Center, Kyushu University, Motooka 744, Nishi-ku, Fukuoka 819-0395, Japan. [5] Japan Synchrotron Radiation Research Insitute (JASRI), SPring-8, 1-1-1 Kouto, Sayo-cho, Sayo-gun, Hyogo 679-5198, Japan. [6] Department of Physical Science, Graduate School of Science, Osaka Prefecture University, Sakai, Osaka 599-8531, Japan. Correspondence and requests for materials should be addressed to K.K. (email: kusada@kuchem.kyoto-u.ac.jp) or to H.K. (email: kitagawa@kuchem.kyoto-u.ac.jp)

The crystal structure is one of the most dominant factors that strongly affect the properties of an alloy, because the electronic structure changes drastically with the crystal structure[1,2]. A solid-solution alloy, in which the constituents are randomly mixed at the atomic scale, generally adopts one of three principal crystal lattice forms: body-centred cubic (bcc), hexagonal close-packed (hcp) and face-centred cubic (fcc) structures. However, once its constituent element and composition are fixed, the crystal structure of solid-solution alloy is uniquely determined[3–5]. Therefore, it is difficult to change the crystal structure of a solid-solution alloy at a certain composition.

Very recently, the crystal structure control of monometallic nanoparticles (NPs) at mild conditions has been reported[6–10]. For example, although bulk gold (Au) adopts only a fcc structure, hcp Au was obtained as an ultrathin nanosheet with a thickness of a few nanometres[8,9]. Furthermore, ruthenium (Ru) NPs with a fcc structure were discovered and exhibited enhanced catalytic properties, even though Ru usually adopts only an hcp structure[10]. These reports opened a new way to design novel monometallic NPs. However, to date, a rational approach to control the crystal structure of metals is not well developed. Given this situation, the rational control of the crystal structure of an alloy system is also very attractive and challenging.

Here, we propose an approach for the selective control of the crystal structure of solid-solution alloys at a certain composition using the chemical reduction method. If the alloy consists of elements whose bulk metals adopt different crystal structures, such as fcc and hcp, the fcc or hcp structure is thermodynamically

and uniquely determined by its composition. However, for NPs, since the crystal structure would be strongly governed by the crystal nucleus during the alloy formation process, it is possible to selectively control its crystal structure by modifying the conditions of the crystal formation process. The crystal structure of binary alloy NPs could be dominated by the structure of the nuclei that are formed from one of the constituent metal ions, which start to be reduced slightly earlier than the other during the alloy formation process. By finely tuning the reduction speed of the metal precursors, which is one of the advantages of the chemical reduction method, we achieve the selective control of the crystal structure. In this paper, a Au–Ru system whose parent metals, Au and Ru, adopt the fcc and hcp structures, respectively, is chosen as the target demonstration for the selective control of the crystal structure, even though they cannot form a solid-solution phase, even in the liquid phase[11]. We succeed in selectively synthesizing fcc and hcp AuRu solid-solution alloy NPs at the Au/Ru ratio of 1:3. The structures of the obtained NPs are investigated using synchrotron powder X-ray diffraction (XRD) analysis and atomic resolution scanning transmission electron microscopy (STEM). In addition, we discuss the mechanism of crystal structure control using the results of ultraviolet–visible (UV–vis) spectral and electrochemical analyses.

## Results

**Syntheses of AuRu$_3$ NPs with fcc and hcp structures.** The AuRu$_3$ solid-solution NPs were synthesized using a polyol reduction method. For the synthesis of the fcc alloy NPs,

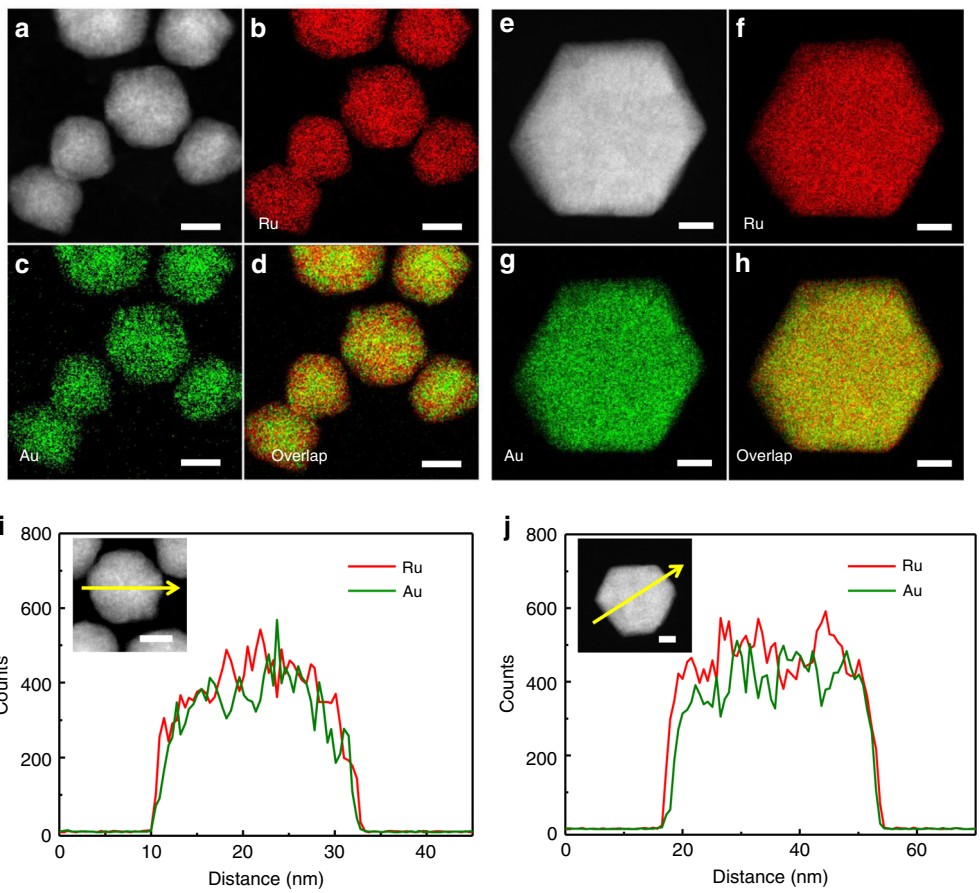

**Fig. 1** EDX maps and line profiles of the synthesized AuRu$_3$ NPs. **a** HAADF-STEM image of fcc-AuRu$_3$ NPs. **b** Ru-L STEM−EDX map of (**a**). **c** Au-M STEM−EDX map of (**a**). **d** Overlay image of (**b**) and (**c**). **e** HAADF-STEM image of an hcp-AuRu$_3$ NP. **f** Ru-L STEM−EDX map of (**e**). **g** Au-M STEM−EDX map of (**e**). **h** Overlay image of (**f**) and (**g**). **i**, **j** EDX line profiles of an fcc-AuRu$_3$ NP and an hcp-AuRu$_3$ NP across the NPs along the arrows shown in the inset figure. Au and Ru are indicated as green and red lines, respectively. All of the scale bars shown in (**a**–**j**) are 10 nm

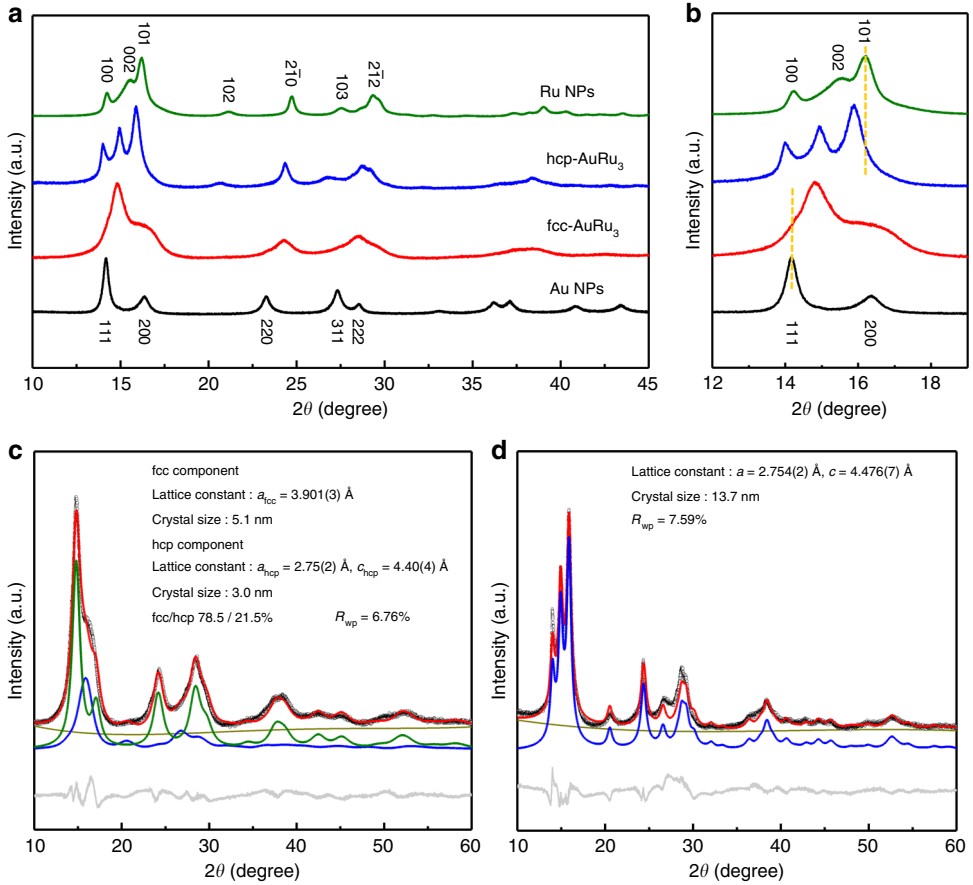

**Fig. 2** Synchrotron XRD analysis of AuRu$_3$ NPs. **a** Synchrotron XRD patterns of Au, fcc-AuRu$_3$, hcp-AuRu$_3$ and Ru NPs at 303 K, $2\theta = 10°$ to 45°. **b** The close-up view of $2\theta = 12°$ to 19°. The radiation wavelength was 0.57865(1) Å. **c**, **d** The Rietveld refinement for the fcc-AuRu$_3$ and hcp-AuRu$_3$ NPs. The diffraction patterns are shown as black circles. The calculated patterns are shown as red lines. The difference profile, the background profile and the fitting curves of the fcc and hcp components are shown as grey, dark yellow, green and blue lines, respectively

hydrogen tetrabromoaurate (III) hydrate (HAuBr$_4$·nH$_2$O) and potassium pentachloronitrosylruthenate (II) (K$_2$Ru(NO)Cl$_5$)[12] were dissolved in diethylene glycol (DEG) with a 1:3 molar ratio. Then, the metal precursor solution was slowly dropped into an ethylene glycol (EG) solution containing polyvinylpyrrolidone (PVP) at 190 °C. The temperature of the solution was maintained at 190 °C during the dropping process. The NPs were separated by centrifuging after cooling to room temperature, as denoted by the fcc-AuRu$_3$ NPs in the following section. In contrast, the synthesis of the hcp AuRu$_3$ alloy NPs was performed via a polyol reduction method in which HAuBr$_4$·nH$_2$O and ruthenium (III) chloride hydrate (RuCl$_3$·nH$_2$O) were used as metal precursors. DEG and PVP were used as a reductant and a protective agent, respectively. In addition, to prepare AuRu$_3$ alloy NPs with an hcp structure, cetyltrimethylammonium bromide (CTAB) was added to adjust the reduction speed of the metal precursors. HAuBr$_4$, RuCl$_3$ and CTAB were dissolved in DEG in a 1:3:20 molar ratio. Then, the metal precursor solution was slowly dropped into a DEG solution containing PVP and CTAB at 215 °C. The NPs were separated by centrifuging after cooling to room temperature, as denoted by the hcp-AuRu$_3$ NPs in the following section.

**Structural analysis**. Transmission electron microscopy (TEM) images of the synthesized AuRu$_3$ NPs were recorded using a Hitachi HT7700 TEM instrument at 100 kV (Supplementary Fig. 1a, c). From the TEM images, the mean diameters of NPs were determined to be 15.8 ± 2.9 nm for fcc-AuRu$_3$ NPs and 85.2

± 6.6 nm for hcp-AuRu$_3$ NPs. The mean diameters were estimated by averaging more than 500 particles (Supplementary Figs. 1b, d).

STEM-energy-dispersive X-ray (EDX) analyses were performed to obtain the direct evidence of the formation of a solid-solution structure. Figure 1a–d and e–h shows high-angle annular dark-field (HAADF)-STEM images, the corresponding Ru-L and Au-M STEM-EDX maps and overlays of Au and Ru maps of fcc-AuRu$_3$ and hcp-AuRu$_3$. In addition, low-magnification EDX maps of the synthesized fcc-AuRu$_3$ and hcp-AuRu$_3$ NPs are shown in Supplementary Figs. 2 and 3. These maps clearly show that Au and Ru atoms are randomly and homogeneously distributed in each NP of fcc-AuRu$_3$ and hcp-AuRu$_3$, although these two elements cannot be mixed with each other, even in the liquid phase in the bulk state. In addition, from the EDX results, the ratios of Au to Ru in fcc-AuRu$_3$ and hcp-AuRu$_3$ are 0.26:0.74 and 0.23:0.77, respectively, which are equal to the nominal ratio in the syntheses. The metal composition of the synthesized NPs was also analysed by using X-ray fluorescence spectroscopy (XRF). The ratios of Au to Ru in fcc-AuRu$_3$ and hcp-AuRu$_3$ are 0.25:0.75 and 0.24:0.76, respectively, which are consistent with the EDX results. We further characterized AuRu$_3$ NPs via the EDX line scanning analysis (Fig. 1i, j). The direction of the line scan is denoted by an arrow across the NP in Fig. 1i, j. These results also show that Au and Ru atoms are homogeneously distributed over the whole particles. These results indicate the formation of AuRu$_3$ solid-solution alloy NPs.

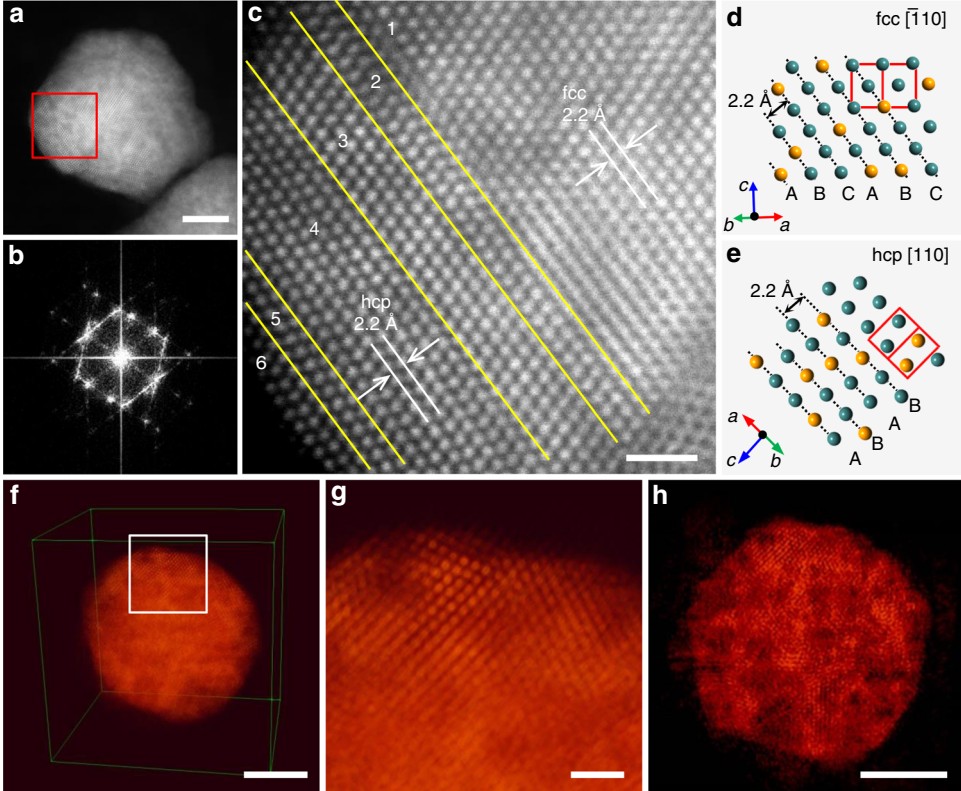

**Fig. 3** STEM analysis and crystal models of fcc-AuRu₃ NPs. **a** Low-magnification HAADF-STEM image of the fcc-AuRu₃ NP. **b** The corresponding FFT pattern of the NP in (**a**). **c** The atomic resolution HAADF-STEM image of the red square region in (**a**). **d** Model for the atomic arrangement of fcc structure viewed along the [$\bar{1}$10] direction. The unit cell is shown as the red frame. **e** Model for the atomic arrangement of the hcp structure viewed along the [110] direction. The unit cell is shown as the red frame. **f** The tomographic reconstruction of the fcc-AuRu₃ NP by EST reconstruction algorithm. **g** Magnified image of the white square region in (**f**). **h** The 0.8 Å-thick slice image of the 3D reconstruction near the centre of the NP. The scale bars in (**a**), (**c**), (**f**), (**g**) and (**h**) are 5, 1, 5, 1 and 5 nm

To investigate the crystal structures of the obtained AuRu₃ solid-solution NPs, XRD measurements were carried out at 303 K, at the beamline BL02B2, SPring-8[13]. Figure 2a shows the XRD patterns of Au, fcc-AuRu₃, hcp-AuRu₃ and Ru NPs. It is obvious that Au and Ru NPs show fcc and hcp diffraction patterns corresponding to their bulk metals. We found that the diffraction pattern of fcc-AuRu₃ NPs is completely different from that of hcp-AuRu₃ NPs. The diffraction patterns of fcc-AuRu₃ and hcp-AuRu₃ NPs were similar to those of Au and Ru NPs, respectively. However, the peak positions of alloy NPs clearly shifted from those of pure Au or Ru NPs (Fig. 2b). This also indicates the formation of AuRu solid-solution alloy.

To clarify the crystal structures of the obtained NPs, the diffraction profiles were analysed by Rietveld refinement (Fig. 2c, d). The result calculated for fcc-AuRu₃ NPs revealed that the NPs consist of a major fcc phase (78.5%) and a minor hcp phase (21.5%). The lattice constant for the fcc component is 3.901(3) Å, which is smaller than that of Au NPs ($a$ = 4.077(3) Å, Supplementary Fig. 4), and the lattice constants of the hcp component are 2.75(2) and 4.40(4) Å for $a_{hcp}$ and $c_{hcp}$, which are larger than those of Ru NPs ($a$ = 2.709(1) Å, $c$ = 4.310(6) Å, Supplementary Fig. 5). The lattice parameter $a_{fcc}$ in a fcc structure is approximately $\sqrt{2}$ $a_{hcp}$ in an hcp structure because both the fcc and hcp structures are close-packed structures. Assuming that the lattice constant follows Vegard's law[14], the Au/Ru atomic ratios of the fcc and hcp phases are calculated to be 0.28:0.72 and 0.24:0.76, which are almost the same values as those found in the EDX analysis. Note that the metal composition of the fcc and hcp phases in fcc-AuRu₃ is equal and that the fcc phase ratio is greater than 75%, even though there are two phases in the synthesized NPs. These results clarified that the AuRu₃ solid-solution NPs with the fcc-phase-dominated crystal structure were successfully prepared.

In contrast, the diffraction profile of hcp-AuRu₃ NPs could be fitted by only an hcp component (Fig. 2d). The lattice constants of the hcp component are 2.754(2) Å for $a$ and 4.476(7) Å for $c$ which are larger than that of Ru, and they are almost consistent with the lattice constants of the hcp component in fcc-AuRu₃ alloy NPs. From Vegard's law, the Au/Ru atomic ratio of hcp-AuRu₃ NPs is calculated to be 0.26:0.74, which is also in accordance with the EDX analysis result. These results confirmed the formation of the solid-solution hcp AuRu₃ alloy NPs. Thus, we successfully showed the first example of selectively controlling the crystal structure of solid-solution alloy in an immiscible Au−Ru system.

We further explored the crystal structure of fcc-AuRu₃ and hcp-AuRu₃ NPs using atomic resolution STEM. Figure 3a, b shows an HAADF-STEM image and a relevant fast Fourier transform (FFT) pattern obtained from the fcc-AuRu₃ particle shown in Fig. 3a. These present a polycrystalline nature of the alloy NP, which is consistent with the fact that the crystal size calculated from the XRD pattern of fcc-AuRu₃ is smaller than the average particle size observed from TEM. Figure 3c shows a clear atomic arrangement of the red area in Fig. 3a. The yellow lines indicate grain boundaries in the particle. The typical atomic arrangement of the fcc phase (ABCABC… stacking sequence of closed packed planes) was observed in the odd-numbered regions, whereas the typical atomic arrangement of the hcp phase

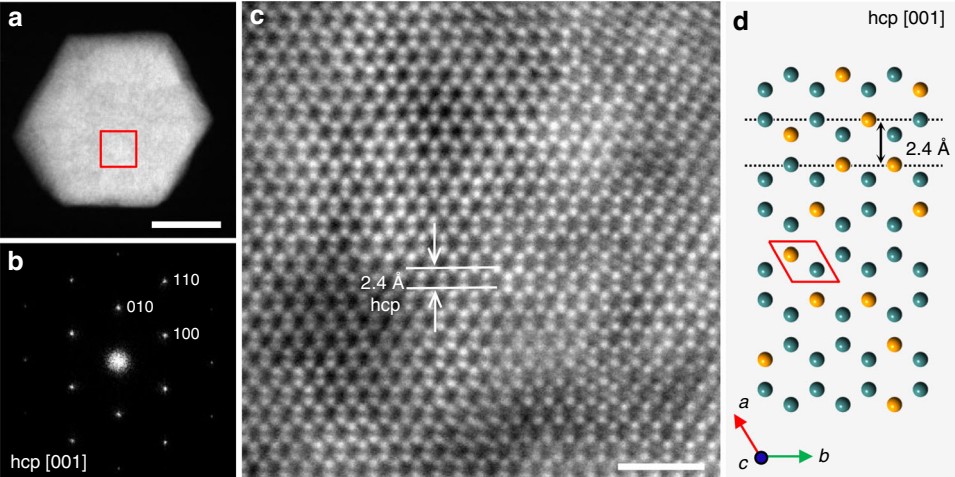

**Fig. 4** STEM analysis and crystal model of hcp-AuRu$_3$ NPs. **a** Low-magnification HAADF-STEM image of an hcp-AuRu$_3$ NP. **b** The corresponding FFT pattern of the NP in (**a**). **c** Atomic resolution HAADF-STEM image of the red square region in (**a**). **d** Model for the atomic arrangement of the hcp structure viewed along the [001] direction. The unit cell is shown as the red frame. The scale bars in (**a**) and (**c**) are 20 and 1 nm

(ABAB…stacking sequence of closed packed planes) was observed in the even-numbered regions. The FFT patterns obtained from the selected areas in Fig. 3c also confirmed the fcc and hcp phases (Supplementary Fig. 6). The FFT pattern obtained from region A in Supplementary Fig. 6 fits the fcc crystal structure with a symmetry of Fm$\bar{3}$ m, as viewed along the [$\bar{1}$ 10] direction. The FFT pattern obtained from region B in Supplementary Fig. 6 fits the hcp crystal structure with a symmetry of P6$_3$/mmc, as viewed along the [110] direction. For further comparison and confirmation, the atomic arrangements of the fcc structure along the [$\bar{1}$ 10] direction and the hcp structure along the [110] direction were simulated (Fig. 3d, e) and are consistent with the atomic arrangements obtained experimentally. In addition, the lattice spacings of both the fcc and hcp phases are observed to be 2.2 Å, which are consistent with the values given via the Rietveld refinement. However, as seen in Fig. 3c, most of the particles consist of the fcc phase, and hcp phases are observed as a fraction of grains between fcc phases like stacking faults.

To investigate the three-dimensional (3D) arrangement of atoms in fcc-AuRu$_3$ NPs, we performed atomic resolution electron tomography with an equally sloped tomography (EST) iterative reconstruction algorithm[15–18] (Fig. 3f–h, Supplementary movies 1 and 2). In the tomographic reconstruction, some grains are observed inside the NP, thus showing the polycrystalline nature of the NP. In addition, the random distribution of both atoms over the whole NP is observed because brighter and darker atoms indicate Au and Ru atoms, respectively, which confirms the solid-solution structure of the NP. Inside the particle, some brighter and darker regions were observed in several atomic-scale regions and were attributed to Au- and Ru-rich regions, respectively. These regions were randomly distributed in both the fcc and hcp grains in three dimensions, and they were smaller than the grain size. This may be derived from the originally immiscible nature of the Au–Ru system.

The crystal structure of hcp-AuRu$_3$ NPs was also investigated using an atomic resolution STEM. Figure 4a, b shows an HAADF-STEM image and a relevant FFT pattern obtained from the particle shown in Fig. 4a. In contrast to fcc-AuRu$_3$ NPs, the FFT pattern in Fig. 4b fits a pure hcp crystal structure with a symmetry of P6$_3$/mmc, viewed along the [001] direction. This presents a single crystalline nature of the alloy NP. Figure 4c shows a clear atomic arrangement of a typical hcp structure viewed along the [001] direction, which is consistent with the

simulated model drawn in Fig. 4d. In addition, the lattice spacing of the (100) plane is observed as 2.4 Å, which is consistent with the Rietveld refinement result. Moreover, some bright and dark areas are observed in several atomic-scale regions as well as in the fcc-AuRu$_3$ NP, while Fig. 4c shows a single hcp arrangement.

**Mechanism of selective control of the crystal structure**. We hypothesized that the selective control of crystal structure in solid-solution alloy NPs can be achieved by finely tuning the reduction speeds of metal precursors with a very small difference. In general, simultaneous reduction of metal precursors is necessary for the formation of solid-solution alloy NPs[19–22]. Otherwise, phase-separated NPs, such as core–shell or segregated types, will be obtained[23]. However, if the reduction occurs completely simultaneously, it may be difficult to control the crystal structure of the solid-solution alloy. We believe that a very small difference in reduction speeds of precursors is needed for the phase control. The crystal structure of binary alloy NPs could be governed by the structure of nuclei made of one of the constituent metal ions, which starts to be reduced slightly earlier than the other during the alloy formation process. Thus, in the target system, if the reduction of Au precursor starts earlier than that of Ru precursor, alloy NPs would adopt a fcc structure since Au naturally forms a fcc structure, as shown in Fig. 5a. While, if the reduction of the Ru precursor starts earlier than that of the Au precursor, the alloy NPs would form an hcp structure because Ru favours an hcp structure.

To prove this hypothesis, the reduction process of Au and Ru precursors was investigated using UV–vis spectroscopy and cyclic voltammetry (CV). To compare the reduction speed of the metal precursors during the formation of alloy NPs, we reduced each precursor under the same conditions used for the syntheses of alloy NPs and monitored the colour change of the solutions that accompanied the reduction of precursors by using UV–vis spectroscopy (Fig. 5b–e).

The fcc AuRu solid-solution alloy NPs were obtained by the reduction of HAuBr$_4$ and K$_2$Ru(NO)Cl$_5$. In the spectra of solution (i) (HAuBr$_4$·nH$_2$O in EG), as shown in Fig. 5b, a peak of [AuBr$_4$]$^-$ at approximately 400 nm, which was assigned to ligand-to-metal charge transfer[24–26], disappeared in 10 s. A peak at approximately 535 nm, which was attributed to typical surface plasmon absorption of Au NPs[27–29], appeared after 10 s, and the intensity of the plasmon peak was saturated at 90 s. In addition, in

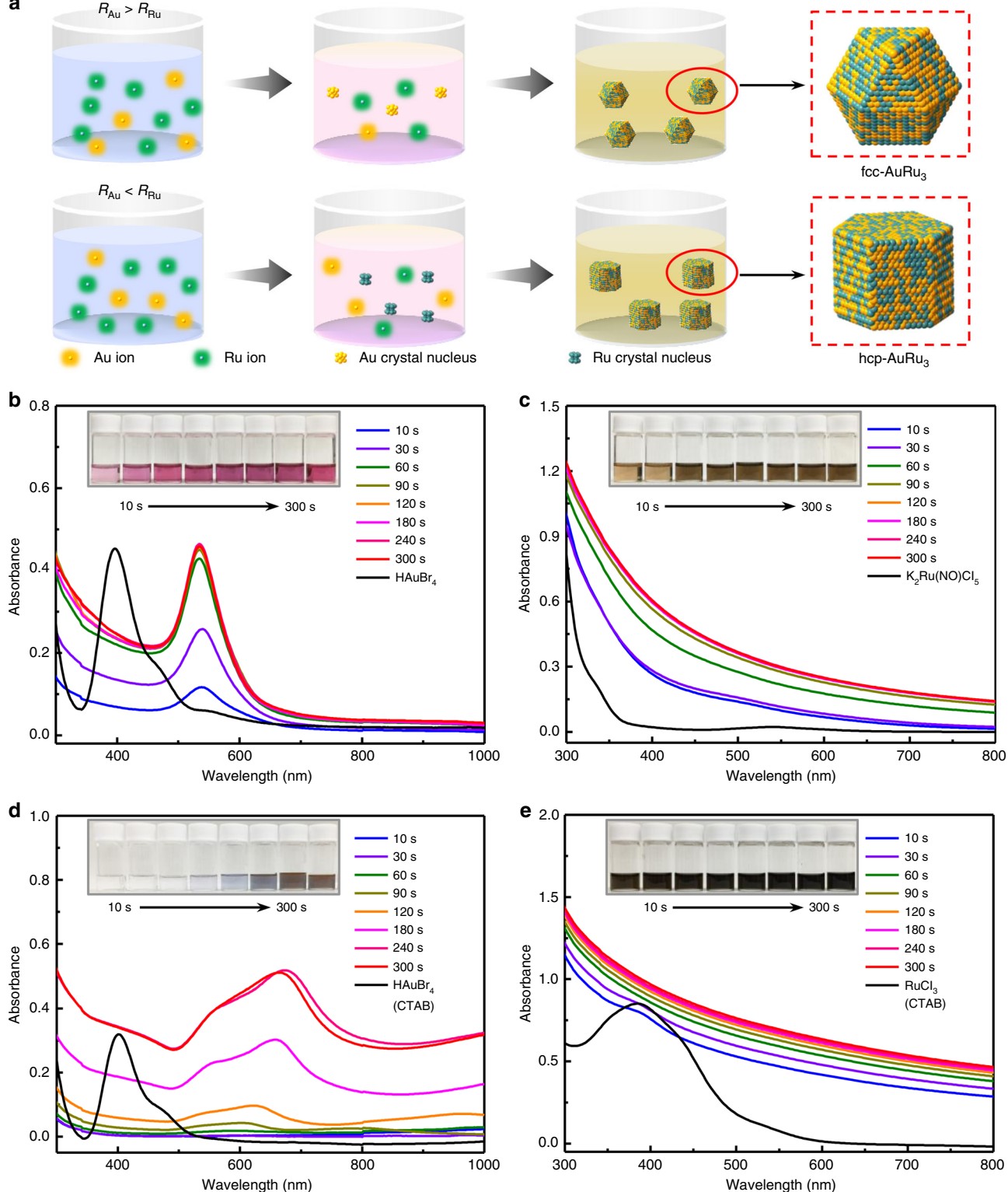

**Fig. 5** Investigation of precursors' reduction speeds. **a** Schematic illustration of the formation process of AuRu$_3$ alloy NPs with fcc and hcp crystal structures. ($R_{Au}$ and $R_{Ru}$ are the reduction speed of the Au and Ru precursors, respectively.) **b**, **c** The change in UV–vis spectra of **b** HAuBr$_4$ and **c** K$_2$Ru(NO)Cl$_5$ in EG with reaction time. **d**, **e** The change in UV–vis spectra of **d** HAuBr$_4$ and **e** RuCl$_3$ with CTAB in DEG with reaction time. The inset pictures in (**b**), (**c**), (**d**) and (**e**) are the samples which were taken at different times

the spectra of solution (ii) ($K_2Ru(NO)Cl_5$ in EG) (Fig. 5c), a broad peak of $[Ru(NO)Cl_5]^{2-}$ at approximately 545 nm disappeared, and another very broad peak at approximately 500 nm appeared after 10 s, which was derived from the $Ru^{2+}/NO^+$ precursor[30]. This broad peak gradually disappeared in 60 s, and the colour of solution turned black, thus implying the formation of Ru NPs at the same time. The spectra did not change after 90 s and are similar to that of the Ru NP solution (Supplementary Fig. 8). By comparing these results, it is suggested that the Au precursor ($HAuBr_4$) is reduced slightly faster than the Ru precursor ($K_2Ru(NO)Cl_5$) under the same conditions used for the synthesis of fcc $AuRu_3$ NPs.

In contrast, hcp AuRu solid-solution alloy NPs were synthesized via the reduction of $HAuBr_4$ and $RuCl_3$ with CTAB. In the spectra of solution (iii) ($HAuBr_4 \cdot nH_2O$ with CTAB in DEG) (Fig. 5d), the peak of $[AuBr_4]^-$ at approximately 400 nm disappeared in 10 s and became colourless. Then, absorption peaks at approximately 500–700 nm, which were attributed to the surface plasmon absorption of Au NPs, appeared after 90 s, and a redshift of the peak at approximately 600 nm was observed with increasing particle size[31-33] (Supplementary Fig. 9). In addition, in the spectra of solution (iv) ($RuCl_3 \cdot nH_2O$ with CTAB in DEG) (Fig. 5e), a broad peak of $RuCl_3$ at approximately 400 nm[34-36] gradually disappeared in 60 s. No peak was observed after 60 s, and the spectra were similar to that of the Ru NP solution (Supplementary Fig. 8). By comparing these results, it is suggested that the Ru precursor ($RuCl_3$) is reduced slightly faster than the Au precursor ($HAuBr_4$) under the same conditions used for the synthesis of hcp $AuRu_3$ NPs.

In addition, CTAB plays a very important role in tuning the reduction speed of the Au precursor because the hcp alloy NPs cannot be obtained without CTAB (Supplementary Figs. 10 and 11). Therefore, we compared the difference between the UV–vis spectra of Au precursors with and without CTAB (Fig. 5d and Supplementary Fig. 12). Without CTAB, Au plasmon absorption peaks were observed at 10 s, while those peaks were observed at 90 s in Fig. 5d. We further investigated the effect of CTAB on the reduction speed of Au by CV (Supplementary Fig. 13). The reduction potential of $[AuBr_4]^-$ without CTAB was measured to be 0.856 V, which is almost in accordance with the value given in the literature[37-39]. However, the reduction potential of $[AuBr_4]^-$ with CTAB was 0.768 V, which was lower than that of $[AuBr_4]^-$ without CTAB. From these results, it is confirmed that the reduction speed of $[AuBr_4]^-$ can be tuned by CTAB. This is probably caused by a complex formation between the Au precursor and CTAB[40,41]. These investigations proved our hypothesis that fine-tuning the reduction speed of metal precursors with a very small difference can realize the selective control of the crystal structure in solid-solution alloy NPs.

## Discussion

Recently, the selective control of the crystal structure in monometallic NPs has received significant attention as an attractive strategy to control their properties. The selective control of the crystal structure in bimetallic NPs at an arbitrary compositional ratio is highly desired to create more effective materials. In this study, we proposed an approach for the selective control of the crystal structure in solid-solution alloys at the same metal composition by finely tuning the reduction speed of the metal precursors with a very small difference, and we demonstrated an example of selective control by synthesizing fcc and hcp $AuRu_3$ alloy NPs. Fine-tuning the reduction speed of the metal precursors was achieved by selecting the appropriate precursors and using CTAB. The combination of $HAuBr_4$ and $K_2Ru(NO)Cl_5$ in which the Au precursor starts to be reduced slightly earlier

provided a fcc structure, while the combination of $HAuBr_4$ and $RuCl_3$ with CTAB, in which the Ru precursor begins to be reduced slightly earlier, provided an hcp structure.

In general, the properties of alloy NPs have been controlled by the size, morphology, constituent elements and compositional ratio. In this work, we introduced a basic material design degree of freedom "crystal structure" to create innovative chemical and physical properties for alloy NPs. Furthermore, this concept can provide a method for controlling the crystal structure of not only Au–Ru system but also other alloy systems consisting of several elements that adopt different structures for the development of innovative electronic, optical, magnetic and catalytic materials.

## Methods

**Preparation of fcc-$AuRu_3$ NPs.** The fcc alloy NPs were prepared using a polyol reduction method. First, PVP (444 mg, molecular weight ≈ 40 000, Wako) was dissolved in EG (100 mL, Wako). Then, the solution was heated to 190 °C with magnetic stirring. Following this, the precursor solution, which was prepared by dissolving hydrogen tetrabromoaurate (III) hydrate ($HAuBr_4 \cdot nH_2O$, 15.2 mg, Alfa Aesar) and potassium pentachloronitrosylruthenate (II) ($K_2Ru(NO)Cl_5$, 29.0 mg, Aldrich) in 10 mL of DEG, was dropped into the hot EG solution at a rate of 1.5 mL min$^{-1}$ using a syringe infusion pump (KDS 200). The temperature of the solution was maintained at approximately 190 °C during the dropping process. After the dropping process, the solution was kept at the same temperature for another 10 min. Then, the NPs were separated via centrifuging after cooling to room temperature.

**Preparation of hcp-$AuRu_3$ NPs.** PVP (444 mg) and CTAB (364.4 mg, Tokyo Chemical Industry Co., Ltd) were dissolved in 100 mL of DEG, and the solution was heated to 215 °C with magnetic stirring. Then, the precursor solution, which was prepared by dissolving $HAuBr_4 \cdot nH_2O$ (15.2 mg, Alfa Aesar), ruthenium (III) chloride hydrate ($RuCl_3 \cdot nH_2O$, 19.6 mg, Wako) and CTAB (182.2 mg) in 10 mL of DEG, was dropped into the hot DEG solution with a speed of 1.5 mL min$^{-1}$. The temperature of the solution was maintained at 215 °C during the dropping process. After the dropping process, the solution was kept at the same temperature for another 10 min. Then, the NPs were separated by centrifuging after cooling to room temperature.

**Characterization.** Synchrotron XRD patterns were measured at the BL02B2 beamline, SPring-8 at room temperature. The radiation wavelength was 0.5786 Å. TEM images were acquired using a Hitachi HT7700 operated at 100 kV. The HAADF-STEM images and EDX spectra were captured using a JEOL JEM-ARM200CF STEM instrument operated at 200 kV. XRF was measured using a Rigaku ZSX Primus IV. Absorption spectra of the solutions were recorded using a JascoV-570 UV–vis spectrophotometer.

**Rietveld refinement.** The Rietveld refinements were performed by using TOPAS3 software developed by Bruker AXS GmbH. The full width at half maximum of silicon (Standard Reference Material 640c) 111 diffraction peak is about 0.015° at $2\theta = 10°$ which is an instrumental resolution function of the diffractometer. All XRD patterns were refined in $2\theta$ range from 10 to 60° with 0.006° per step identical to the resolution of the beamline BL02B2, SPring-8. The XRD pattern of fcc-$AuRu_3$ was well fitted by two components with space groups of $Fm\overline{3}m$ and $P6_3/mmc$. For hcp-$AuRu_3$, the pattern was fitted by one component of $P6_3/mmc$.

**HAADF-STEM tomography.** The tilt series of HAADF-STEM images for the individual $AuRu_3$ NP was acquired at the angle range from −55.6° to 65.6° using JEM-ARM200F operated at 120 kV (Supplementary Fig. 7). The image size was 1024 × 1024 pixels at a pixel size of 0.20 × 0.20 Å$^2$. To reduce electron beam damage, 10 images were acquired using a small pixel time of 1 μs px$^{-1}$ and a probe current of 12 pA at each angle step. The image series at each angle step were averaged, and the image size was reduced to 512 × 512 pixels, after image alignment and affine-transforms to correct image distortion due to specimen drifts. The intensity of the averaged tilt image series was normalized using the intensity of a carbon film part to correct the intensity variation due to the electron probe current changes. The background of each projection was removed after alignment using the cross-correlation method[42]. The EST reconstruction algorithm[15-17] developed by the UCLA group and their codes were used for 3D tomographic reconstruction. In EST reconstruction, Fourier-space-based iterative reconstructions were performed with real-space constrains to minimize the difference between observed images and the reconstruction projections using a pseudo polar fast Fourier transform (PPFFT) and the inverse PPFFT with oversampling[15-17]. The detailed scheme of the EST reconstruction algorism was described in the supporting information of ref. [15]. In this study, the 500 times iteration was carried out to minimize the R-factor and two

times oversampling PPFFT were used. The reconstructed data were visualized via the Visualizer Kai post processing software (Systems In Frontiers Inc.).

**Reduction speed measurements**. To compare the reduction speed of metal precursors during the formation of alloy NPs, each precursor was reduced under the same conditions used for the syntheses of alloy NPs and detected the colour change of the solutions accompanying the reduction of precursors via the UV–vis spectroscopy. The measurements for each metal precursor were repeated at least twice. In a standard procedure, the metal precursor used for the formation of fcc-$AuRu_3$ NPs, $K_2Ru(NO)Cl_5$ (0.075 mmol) or $HAuBr_4$ (0.025 mmol), was first dissolved in 10 mL of EG. Then, 100 mL of the EG solvent was heated to 195 °C with stirring. Then, the precursor solvent was quickly injected into the hot EG solvent and was heated at a temperature of approximately 190 °C. During the heating process, 1.5 mL aliquots were sampled from the reaction solution using a glass pipet and immediately immersed into an ice bath to quickly quench the reduction reaction. The samples were taken at 10, 30, 60, 90, 120, 180, 240 and 300 s and diluted to 3 mL by using EG. The absorption spectra of samples were measured using a UV–vis spectrophotometer. The same procedure was performed to investigate the reduction process of metal precursors used in the formation of hcp-$AuRu_3$ NPs. The differences were that DEG was used as a solvent instead of EG and CTAB (1.5 mmol) was added to the solvent. The temperature was kept at approximately 215 °C.

**Electrochemical measurements**. The concentrations of $HAuBr_4$ and CTAB were 0.5 mmol $L^{-1}$ and 10 mmol $L^{-1}$. A CHI 760E electrochemical analyser (CH Instruments) was used for collecting the electrochemical data. All of the experiments were performed at room temperature using a conventional three-electrode system with a 3.0 mm diameter glassy carbon disk working electrode, a platinum wire auxiliary electrode and a Ag/AgCl (3 M NaCl) reference electrode. All of the electrodes were purchased from ALS Co., Ltd. The working electrode surface was polished with 1 μM diamond and 0.05 μM alumina solutions (ALS Co., Ltd) before the tests. The measurements were carried out in an Ar-saturated 0.05 M $H_2SO_4$ aqueous solution. During the measurement, the atmosphere above the solution was kept inert with a constant flow of Ar. The cyclic voltammograms were recorded at a scan rate of 0.02 V $s^{-1}$ and measured over the potential range of +0.7 to −0.1 V (vs. Ag/AgCl).

**Data availability**. The authors declare that the data supporting the findings of this study are available within the paper and its supplementary information files.

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

## Acknowledgements

This research was supported by the ACCEL program, Japan Science and Technology Agency (JST), JPMJAC1501. STEM observations were performed as part of a program conducted by the Advanced Characterization Nanotechnology Platform sponsored by the MEXT of the Japanese Government. Synchrotron XRD measurements were carried out at SPring-8 under proposal No. 2014B1382, 2015A1586 and 2016A1483. The activities of the INAMORI Frontier Research Center, Kyushu University are supported by KYOCERA Corporation.

## Author contributions

K.K. and H.K. conceived and designed the research. Q.Z. performed the experiments. T.Y., T. T. and S.M. performed the HAADF-STEM and EDX analyses. S.K. and Y.K. contributed to the synchrotron XRD measurements. Q.Z. and D.W. performed electrochemical experiments. Q.Z., K.K. and H.K. analysed and discussed the experimental results and wrote the manuscript. All of the authors discussed and commented on the paper.

## Additional information

**Competing interests:** The authors declare no competing financial interests.

