## [Peer Review File · Nature Communications]

Reviewers' comments:

Reviewer #1 (Remarks to the Author):

The paper described new chemical procedures to selective control the structure of alloy nanoparticles. As the structure is such an important factor in determining the overall properties, this constitutes substantial progress. The paper is well written and experimental work is carried out well and justifies the claims. I would suggest the authors spend a little more time on discussing the wider implications of their results, and how universal these techniques could be for other systems, but otherwise I would recommend the paper for publication.

Reviewer #2 (Remarks to the Author):

Crystal structure is one of the most important factors that can significantly affect the properties of the inorganic materials just like the size and morphology. At present, the property control of alloy NPs was achieved mainly through size and morphology adjustment. The property modulation of alloy NPs by tuning their crystal structure is desired but still a big challenge. In this manuscript, the authors reported a new chemical reduction method to selectively control the crystal structure of solid-solution alloys. By precisely controlling the reduction speed of the metal precursors, AuRu₃ alloy NPs with both fcc and hcp structure are successfully synthesized, respectively. This bottom-up controlling method will bring other researchers a different thinking way to create novel chemical and physical properties for alloy NPs. And therefore, it would attract great attentions from other researchers in the fields like chemistry, physics, material science and so on. This paper has been carefully organized with detailed experimental data and appropriate discussion. So, I think this work has excellent novelty and suggest accepting it after following minor revisions:

1 In the manuscript, the authors confirmed the composition of the AuRu₃ alloy NPs from the EDX analysis. However, since the instrumental error may exist for different EDX analysis equipment, other characterization methods need to be employed to further confirm the composition of the alloy NPs, for example, XRF or ICP measurement.

2 The Rietveld refinement was used to analysis the XRD pattern of the alloy NPs in this work. But, the authors didn't mention any details about this calculation. It is important to provide more details about the calculation. For example, what kind of software was used? How about the calculation step values

3 It is very interesting to reconstruct the three-dimensional arrangement of atoms in AuRu₃ NPs. Since this is a very new analysis method and few researchers can easily understand it well, more details about the method of tomographic reconstruction need to be described in the method part.

4 The authors provide the three-dimensional arrangement of atoms in fcc-AuRu₃ NPs in the manuscript. For convenient comparison, please provide the three-dimensional arrangement of atoms in hcp-AuRu₃ NPs if possible.

5 During the mechanism exploration part, the authors mentioned that the CTAB will effective reduce the reducing speed of gold precursor. Since the gold and ruthenium precursors were mixed together with CTAB during the synthesis of the AuRu₃ NPs, how did CTAB affect the reduction speed of ruthenium precursor?

6 In this manuscript, the authors reported the crystal structure control at an Au:Ru ratio of 1:3. Since this control was realized by carefully adjusting the reducing speed of the metal precursors, it would be difficult to realize the crystal structure control on other ratios. Is it possible to control the structure on other compositions with the same condition of AuRu₃?

Response sheet of manuscript NCOMMS-17-26928

Comments from Reviewer #1:

The paper described new chemical procedures to selective control the structure of alloy nanoparticles. As the structure is such an important factor in determining the overall properties, this constitutes substantial progress. The paper is well written and experimental work is carried out well and justifies the claims. I would suggest the authors spend a little more time on discussing the wider implications of their results, and how universal these techniques could be for other systems, but otherwise I would recommend the paper for publication.

Answer to comments from Reviewer #1:

Thank you very much for recognizing the significance of our work and giving valuable comments. We believe our approach is universal and we are now applying this new approach to other alloy systems such as Ru(hcp)-Ag(fcc), Mo(bcc)-Pd(fcc) and Fe(bcc)-Co(hcp). We have already achieved some progress; however, we need more efforts to complete these works and hope to keep confidential before being published in near future. However, we added some sentence at the end of the manuscript as possible applications.

Revision 1:

P18, line 2; Furthermore, this concept can provide a new method for controlling the crystal structure of not only Au–Ru system but also other alloy systems consisting of several elements that adopt different structures for the development of innovative electronic, optical, magnetic and catalytic materials.

Comments from Reviewer #2:

Crystal structure is one of the most important factors that can significantly affect the properties of the inorganic materials just like the size and morphology. At present, the property control of alloy NPs was achieved mainly through size and morphology adjustment. The property modulation of alloy NPs by tuning their crystal structure is desired but still a big challenge. In this manuscript, the authors reported a new chemical reduction method to selectively control the crystal structure of solid-solution alloys. By precisely controlling the reduction speed of the metal precursors, AuRu₃ alloy NPs with both fcc and hcp structure are successfully synthesized, respectively. This bottom-up controlling method will bring other researchers a different thinking way to create novel chemical and physical properties for alloy NPs. And therefore, it would attract great attentions from other researchers in the fields like chemistry,

physics, material science and so on. This paper has been carefully organized with detailed experimental data and appropriate discussion. So, I think this work has excellent novelty and suggest accepting it after following minor revisions:

Comment 1:

In the manuscript, the authors confirmed the composition of the AuRu₃ alloy NPs from the EDX analysis. However, since the instrumental error may exist for different EDX analysis equipment, other characterization methods need to be employed to further confirm the composition of the alloy NPs, for example, XRF or ICP measurement.

Answer to comment 1:

We are grateful for your useful comments. According to your comments, we conducted further XRF measurement for both fcc-AuRu₃ and hcp-AuRu₃ NPs. From the XRF results, the ratios of Au to Ru in fcc-AuRu₃ and hcp-AuRu₃ are 0.25:0.75 and 0.24:0.76, respectively, which are exactly the same to the EDX results. We have added the sentences about the XRF results as shown below.

Revision 2:

P5, line 17; The metal composition of the synthesized NPs was also analysed by using X-ray fluorescence spectroscopy (XRF). The ratios of Au to Ru in fcc-AuRu₃ and hcp-AuRu₃ are 0.25:0.75 and 0.24:0.76, respectively, which are consistent with the EDX results.

P19, line 11; X-ray fluorescence spectroscopy (XRF) was measured using a Rigaku ZSX Primus IV.

Comment 2

The Rietveld refinement was used to analysis the XRD pattern of the alloy NPs in this work. But, the authors didn't mention any details about this calculation. It is important to provide more details about the calculation. For example, what kind of software was used? How about the calculation step values

Answer to comment 2:

Thank you very much for your valuable comments. The Rietveld refinements were performed by using TOPAS3 software developed by Bruker AXS GmbH. The full width at half maximum of silicon (Standard Reference Material 640c) 111 diffraction peak is about 0.015° at 2θ = 10° which is an instrumental resolution function of the diffractometer. All XRD patterns were refined in 2θ range from 10 to 60° with 0.006°/step identical to the resolution of the beamline BL02B2, SPring-8. The XRD pattern of fcc-AuRu₃ was well fitted by two components with space groups of *Fm-3m* and *P63/mmc*. For hcp-AuRu₃, the pattern was fitted by only one component of *P63/mmc*. In addition, we have added a reference (Kawaguchi, S. *et al.* High-throughput powder diffraction measurement system consisting of multiple MYTHEN detectors at beamline BL02B2 of SPring-8. *Rev. Sci. Instrum.* **88**, 085111 (2017)) reporting the

details of the beamline. We have added the sentences about the Rietveld refinement in the method part as follows.

Revision 3:

P6, line 1; To investigate the crystal structures of the obtained AuRu₃ solid-solution NPs, XRD measurements were carried out at 303 K, at the beamline BL02B2, SPring-8¹³.

P19, line 14; Rietveld refinement. The Rietveld refinements were performed by using TOPAS3 software developed by Bruker AXS GmbH. The full width at half maximum of silicon (Standard Reference Material 640c) 111 diffraction peak is about 0.015° at 2θ = 10° which is an instrumental resolution function of the diffractometer. All XRD patterns were refined in 2θ range from 10 to 60° with 0.006°/step identical to the resolution of the beamline BL02B2, SPring-8. The XRD pattern of fcc-AuRu₃ was well fitted by two components with space groups of *Fm-3m* and *P63/mmc*. For hcp-AuRu₃, the pattern was fitted by one component of *P63/mmc*.

Comment 3

It is very interesting to reconstruct the three-dimensional arrangement of atoms in AuRu₃ NPs. Since this is a very new analysis method and few researchers can easily understand it well, more details about the method of tomographic reconstruction need to be described in the method part.

Answer to comment 3:

Thank you very much for your careful comment. We have added some details in the method of tomographic reconstruction part as follows.

Revision 4:

P20, line 5; The intensity of the averaged tilt image series was normalized using the intensity of a carbon film part to correct the intensity variation due to the electron probe current changes. The background of each projection was removed after alignment using the cross-correlation method⁴². The Equally Sloped Tomography (EST) reconstruction algorithm¹⁵⁻¹⁷ developed by the UCLA group and their codes were used for 3D tomographic reconstruction. In EST reconstruction, Fourier-space-based iterative reconstructions were performed with real-space constrains to minimize the difference between observed images and the reconstruction projections using a pseudo polar fast Fourier transform (PPFFT) and the inverse PPFFT with over sampling¹⁵⁻¹⁷. The detailed scheme of the EST reconstruction algorithm was described in the supporting information of the reference 15. In this study, the 500 times iteration was carried out to minimize the R-factor and two times oversampling PPFFT were used.

Comment 4

The authors provide the three-dimensional arrangement of atoms in fcc-AuRu₃ NPs in the manuscript. For convenient comparison, please provide the three-dimensional arrangement of atoms in hcp-AuRu₃ NPs if possible.

Answer to comment 4:

Thank you very much for your comments. Unfortunately, the atomic-resolution tomography technique can be only applied to nanoparticles with several nanometer sizes because of a limitation of a narrow depth of field in STEM imaging. In atomic-resolution STEM imaging, convergent electron beams with large convergent angles of 15-30 mrad are often used to obtain atomic resolution, which restricts the depth of fields to several nanometers. Thus, the hcp-AuRu₃ NPs with average size of 85.2 nm are too large for atomic-resolution-tomography. In the case of fcc-AuRu₃ NP, we selected the NP smaller than the average size of 15.8 nm. The selected size around 10 nm is almost maximum size for tomographic reconstruction with atomic resolutions.

Comment 5

During the mechanism exploration part, the authors mentioned that the CTAB will effective reduce the reducing speed of gold precursor. Since the gold and ruthenium precursors were mixed together with CTAB during the synthesis of the AuRu₃ NPs, how did CTAB affect the reduction speed of ruthenium precursor?

Answer to comment 5:

Thank you very much for your careful comment. Hydrogen tetrabromoaurate (III) hydrate (HAuBr₄·nH₂O) and ruthenium (III) chloride hydrate (RuCl₃·nH₂O) were used as metal precursors together with CTAB during the synthesis of hcp-AuRu₃ NPs. Therefore, we compared the reducing speed of the ruthenium precursor with and without CTAB as well the gold precursor discussed in the manuscript. The change in UV-Vis spectra of RuCl₃·nH₂O without CTAB was shown below. We found that the reduction speed of RuCl₃·nH₂O was much less influenced by CTAB than the gold precursor.

Figure. UV-Vis spectra of RuCl₃·nH₂O in DEG with increasing reaction time. Insert photo shows the colour change during this process.

Comment 6

In this manuscript, the authors reported the crystal structure control at an Au:Ru ratio of 1:3. Since this control was realized by carefully adjusting the reducing speed of the metal precursors, it would be difficult to realize the crystal structure control on other ratios. Is it possible to control the structure on other compositions with the same condition of AuRu₃?

Answer to comment 6:

Thank you very much for your useful comments. We are investigating the crystal structure control for other ratios. As you mentioned, this control must be carefully made by adjusting the reducing speed of the metal precursors. Since the reducing condition is slightly different along with the concentration of the precursors, it is very difficult to perfectly control the crystal structure on other ratios by using the completely same reaction condition for AuRu₃. However, very recently, we have partly achieved the control on other compositions such as 1:1 based on this approach. We need more adjustments to report these results as a full paper. We hope that this work will be published in the near future.

REVIEWERS' COMMENTS:

Reviewer #2 (Remarks to the Author):

I am satisfied with the revisions made by the authors. I think this paper could be accepted for publication as it now.

Response sheet of manuscript *NCOMMS-17-26928*

Comments from Reviewer #2:

I am satisfied with the revisions made by the authors. I think this paper could be accepted for publication as it now.

Response to comments (Reviewer #2):

Thank you very much for your comments.